# AttnDreamBooth: Towards Text-Aligned Personalized Text-to-Image Generation

**Lianyu Pang**[1]    **Jian Yin**[1]    **Baoquan Zhao**[1]    **Feize Wu**[1]    **Fu Lee Wang**[2]
**Qing Li**[3]    **Xudong Mao**[1]*
[1]Sun Yat-sen University    [2]Hong Kong Metropolitan University
[3]The Hong Kong Polytechnic University
https://attndreambooth.github.io

## Abstract

Recent advances in text-to-image models have enabled high-quality personalized image synthesis based on user-provided concepts with flexible textual control. In this work, we analyze the limitations of two primary techniques in text-to-image personalization: Textual Inversion and DreamBooth. When integrating the learned concept into new prompts, Textual Inversion tends to overfit the concept, while DreamBooth often overlooks it. We attribute these issues to the incorrect learning of the embedding alignment for the concept. To address this, we introduce AttnDreamBooth, a novel approach that separately learns the embedding alignment, the attention map, and the subject identity across different training stages. We also introduce a cross-attention map regularization term to enhance the learning of the attention map. Our method demonstrates significant improvements in identity preservation and text alignment compared to the baseline methods.

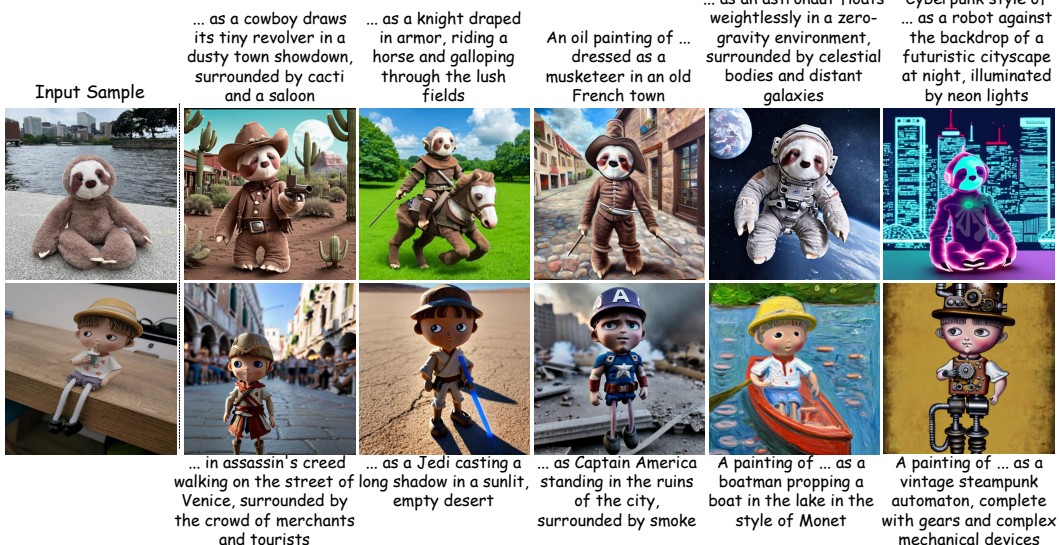

... as a cowboy draws its tiny revolver in a dusty town showdown, surrounded by cacti and a saloon

... as a knight draped in armor, riding a horse and galloping through the lush fields

An oil painting of ... dressed as a musketeer in an old French town

... as an astronaut floats weightlessly in a zero-gravity environment, surrounded by celestial bodies and distant galaxies

Cyberpunk style of ... as a robot against the backdrop of a futuristic cityscape at night, illuminated by neon lights

Input Sample

... in assassin's creed walking on the street of Venice, surrounded by the crowd of merchants and tourists

... as a Jedi casting a long shadow in a sunlit, empty desert

... as Captain America standing in the ruins of the city, surrounded by smoke

A painting of ... as a boatman propping a boat in the lake in the style of Monet

A painting of ... as a vintage steampunk automaton, complete with gears and complex mechanical devices

Figure 1: Our method enables text-aligned text-to-image personalization with complex prompts.

---

*Corresponding author (xudong.xdmao@gmail.com).

38th Conference on Neural Information Processing Systems (NeurIPS 2024).

# 1 Introduction

Text-to-image personalization [24, 71, 50] is the task of customizing a pre-trained diffusion model to produce images of user-provided concepts in novel scenes or styles. By providing several examples of a new concept, personalization techniques enable users to employ novel prompts to generate personalized images containing that concept. Current personalization techniques primarily fall into two categories: the first approach involves inverting the new concept into the textual embedding [24]; the second approach involves fine-tuning the diffusion model to learn the new concept [71]. Personalization techniques aim to generate high-quality images of user-provided concepts, achieving high identity preservation and text alignment. However, despite the significant progress in personalization techniques, balancing the trade-off between identity preservation and text alignment remains a challenge for current approaches.

Figure 2 shows the personalization results from Textual Inversion [24] and DreamBooth [71]. Textual Inversion tends to generate images that focus primarily on the learned concept, often neglecting other elements of the prompt. In contrast, DreamBooth appears to overlook the learned concept, producing images that are more influenced by other prompt tokens. These issues can be attributed to the incorrect learning of embedding alignment for the new concept, i.e., the embedding of the new concept is not functionally compatible with the embeddings of existing tokens.

Based on these observations, our approach aims to properly learn not only the subject identity but also the embedding alignment and the attention map for the new concept. Our key insights are as follows: 1) In the early stages of optimization, Textual Inversion effectively learns the embedding alignment but tends to overfit after extensive optimization steps; 2) DreamBooth accurately captures the subject identity but struggles with learning the embedding alignment.

In this paper, we propose a method named AttnDreamBooth, which separates the learning processes of the embedding alignment, the attention map, and the subject identity. Specifically, our approach consists of three main training stages, as illustrated in Figure 3. First, we optimize the textual embedding to learn the embedding alignment while preventing the risk of overfitting, which results in a coarse attention map for the new concept. Next, we fine-tune the cross-attention layers of the U-Net to refine the attention map. Lastly, we fine-tune the entire U-Net to capture the subject identity. Note that the text encoder remains fixed throughout all training stages to preserve its prior knowledge of contextual understanding.

Furthermore, we introduce a cross-attention map regularization term to enhance the learning of the attention map. Throughout the three training stages, we use a consistent training prompt, "a photo of a [V] [super-category]", where [V] and [super-category] denote the tokens for the new concept and its super-category, respectively. Our attention map regularization term encourages similarity between the attention maps of the new concept and its super-category.

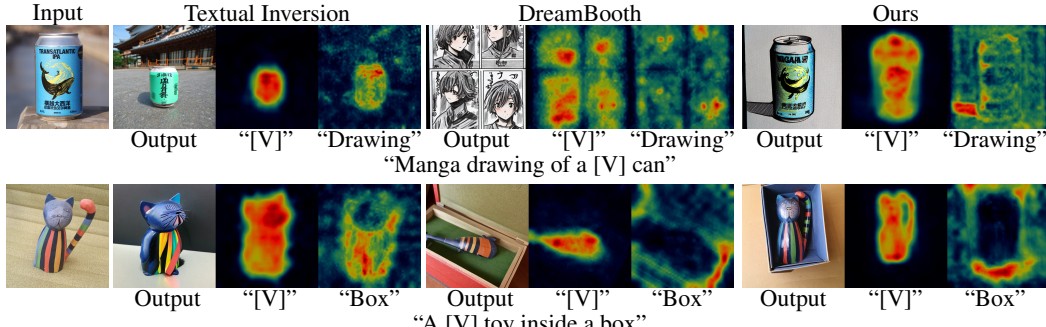

Figure 2: **Analysis of two principal methods.** We visualize the cross-attention maps corresponding to the new concept and other tokens in the prompt. Textual Inversion [24] tends to overfit the textual embedding of the learned concept, resulting in incorrect attention map allocations to other tokens (e.g., "drawing" or "box"). In contrast, DreamBooth [71] appears to overlook the learned concept, producing images primarily based on other tokens.

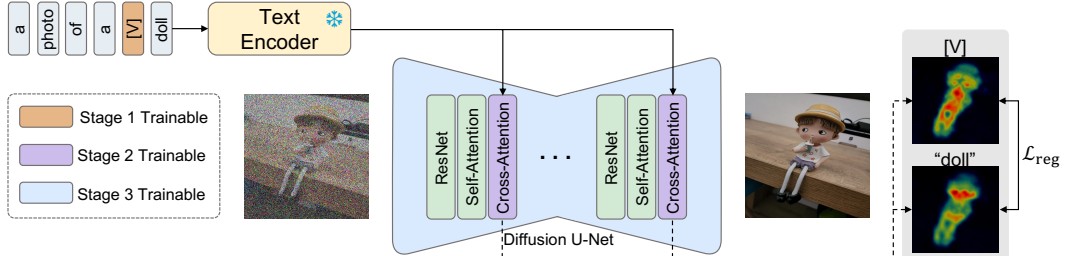

Figure 3: **Overview of AttnDreamBooth**. Our method consists of three training stages. In Stage 1, we optimize the textual embedding of the new concept to align its embedding with existing tokens. In Stage 2, we fine-tune the cross-attention layers to refine the attention map. In Stage 3, we fine-tune the entire U-net to capture the subject identity. Moreover, we introduce a cross-attention map regularization term to guide the learning of the attention map.

To demonstrate the effectiveness of AttnDreamBooth, we compare it with four state-of-the-art baseline methods through both qualitative and quantitative evaluations. Our method achieves superior performance in terms of identity preservation and text alignment compared to the baselines. More importantly, AttnDreamBooth enables a variety of text-aligned personalized generations with complex prompts.

## 2 Related Work

**Text-to-Image Generation.** Generative models are designed to create new samples that resemble the patterns observed in their training data. There are various types of generative models, including VAEs [47, 76, 14], GANs [27, 6, 44], auto-regressive models [66, 91], flow-based models [21, 48], and diffusion models [75, 33, 59, 18]. These models can be enhanced by conditioning on text prompts, which are known as text-to-image models [68, 66, 60, 20, 23, 16, 5]. Recent advancements [73, 67, 70] in text-to-image generation, powered by training on extremely large-scale datasets, have demonstrated an impressive ability to generate diverse and generalized outputs.

**Text-to-Image Personalization.** Leveraging the impressive capabilities of diffusion models, text-to-image personalization involves adapting pre-trained diffusion models to capture new concepts depicted in several given images. Pioneering works approach this by inverting the concept into the textual embedding [24], or by fine-tuning the diffusion model [71]. However, these methods often struggle to balance the trade-off between identity preservation and text alignment, and typically require substantial time for optimization. To overcome these limitations, some studies focus on enhancing the identity preservation of the concept [81, 1, 94, 36, 31, 43, 40], while others aim to improve text alignment [78, 3, 4, 90, 37]. Additionally, there is a growing trend of research attempting to accelerate the personalization process, either by reducing the number of tuning parameters [50, 29, 54, 34, 28, 57], or by pre-training on large datasets [85, 74, 39, 2, 25, 10, 72, 51, 87, 56, 11, 55]. Given the widespread interest in human synthesis, many studies also concentrate on the personalized synthesis of human faces [92, 62, 89, 53, 79, 83, 8, 49, 61, 45, 86, 17, 13, 82, 12].

**Cross-Attention Control.** The cross-attention layers [70] have been shown to play a crucial role in diffusion models. The control of cross-attention layers has proven effective in a variety of tasks, including image editing [32], compositional synthesis [22, 52, 7, 26, 46], and layout-controlled synthesis [63, 9, 88]. In text-to-image personalization, several studies [50, 87, 4, 78, 30, 42, 58, 93] also have explored the control of cross-attention layers. Custom Diffusion [50] illustrates how incorrect attention maps of the learned concepts can lead to unsuccessful synthesis. FastComposer [87] and Break-A-Scene [4] propose using segmentation masks of the target concepts to guide the learning of the attention maps, thereby enhancing text alignment, especially in scenarios involving multiple concepts. Perfusion [78] identifies the attention overfitting issue and addresses it by fixing the cross-attention key matrices of the target concepts to their super-category tokens.

**Multi-Stage Personalization.** Several studies [24, 54, 4, 41, 38] have explored combining the strengths of different methods into more efficient models through a multi-stage approach. Inspired by

PTI [69], Textual Inversion [24] investigated a two-stage approach to enhance identity preservation by first optimizing the textual embedding and then fine-tuning the diffusion model to better capture the subject identity. Similarly, MagiCapture [38] first optimizes the textual embedding and then applies LoRA [34] in the U-Net. Break-A-Scene [4] proposes initially optimizing the textual embedding with a high learning rate, followed by fine-tuning both the U-Net and the text encoder using a significantly lower learning rate. Our method differs in several aspects. First, the motivation for our first stage (i.e., optimizing the textual embedding) differs from previous methods. Specifically, we focus on learning the embedding alignment while mitigating the risk of overfitting, thus significantly reducing the optimization steps and lowering the learning rate. Second, we decompose the learning process into three stages: learning the embedding alignment, refining the attention map, and capturing the subject identity. Third, we introduce a cross-attention map regularization to guide the learning of the attention map in a self-supervised manner.

## 3   Preliminaries

**Latent Diffusion Models.**   Our approach is based on the publicly available Stable Diffusion model, a type of Latent Diffusion Model (LDM) [70] for text-to-image generation. In LDM, an autoencoder is utilized to provide a lower-dimensional representational space, where an encoder $\mathcal{E}$ transforms an image $x$ into a latent representation $z = \mathcal{E}(x)$, and a decoder $\mathcal{D}$ reconstructs the image from this latent code, i.e., $\mathcal{D}(\mathcal{E}(x)) \approx x$. Additionally, a Denoising Diffusion Probabilistic Model (DDPM) [33] is employed to produce latent codes within the latent space of the autoencoder. To generate images from text, the model leverages a conditioning vector $c(y)$, derived from a given text prompt $y$. The training objective of LDM is given by:

$$\mathcal{L}_{\text{diffusion}} = \mathbb{E}_{z \sim \mathcal{E}(x), y, \varepsilon \sim \mathcal{N}(0,1), t} \left[ \| \varepsilon - \varepsilon_\theta \left( z_t, t, c(y) \right) \|_2^2 \right],  \tag{1}$$

where the denoising network $\varepsilon_\theta$ is tasked with recovering the original latent code $z_0$ from the noised latent code $z_t$, given a specific timestep $t$ and the conditioning vector $c(y)$.

**Textual Inversion.**   Textual Inversion (TI) [24] personalizes a pre-trained diffusion model by encoding the target concept into the textual embedding. Given several images of a target concept, TI introduces a new token $S_*$ and its associated textual embedding $v_*$ to represent the concept. The learning process of TI involves initializing $v_*$ with a coarse descriptor and then optimizing it to minimize the diffusion objective (Eq. 1).

**DreamBooth.**   DreamBooth (DB) [71] learns the target concept by fine-tuning the pre-trained diffusion model. Given several images of a target concept, DB labels all the images with the prompt "a [V] [super-category]", where [V] is a rare token in the vocabulary. The learning process of DB involves fine-tuning the entire U-Net (and possibly the text encoder) using the diffusion objective (Eq. 1) combined with a prior preservation loss [71].

## 4   Method

In this section, we first analyze the problems associated with Textual Inversion and DreamBooth, as discussed in Section 4.1. To address these issues, we propose a novel method named AttnDreamBooth, as detailed in Section 4.2. To further enhance text alignment, we introduce a cross-attention map regularization term in Section 4.3.

### 4.1   Analysis of Existing Methods

**Problems and Analysis.**   As illustrated in Figure 2, Textual Inversion and DreamBooth encounter distinct challenges when integrating the learned concept into novel prompts. For Textual Inversion, the generated images often excessively focus on the learned concept, overlooking other prompt tokens. To investigate this issue, we present the attention map visualization using DAAM [77] for different tokens in Figure 2. This visualization reveals an embedding misalignment issue in novel compositions containing the concept, leading to incorrect attention map allocations for other tokens. A typical example is shown where the attention map corresponding to the "drawing" token focuses on incorrect regions. This misalignment occurs because Textual Inversion tends to overfit

| Input | Output | "[V]" | "Monet" | Before fine-tuning | After fine-tuning |
|---|---|---|---|---|---|

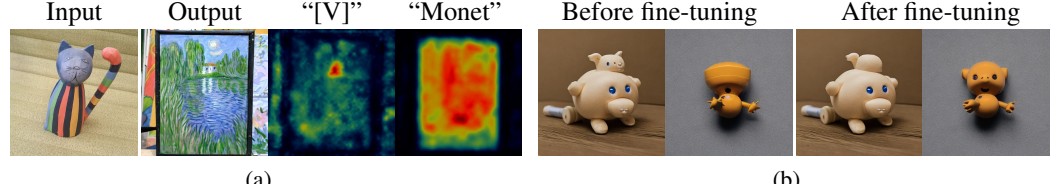

| (a) | (b) |
|---|---|

Figure 4: **Analysis of TI+DB**. Column (a) demonstrates that TI+DB neglects the learned concept when integrating it into a new prompt, "A painting of a [V] toy in the style of Monet". Column (b) shows the generated images based on a single word prompt, "[V]", both before and after fine-tuning, using the diffusion model without fine-tuning. These images are notably similar to each other, which indicates that the learned textual embedding remains largely unchanged from its initial state.

the input embedding of the text encoder, responsible for managing the contextual understanding of the prompt. Conversely, images generated by DreamBooth sometimes focus solely on other prompt tokens, neglecting the learned concept. This occurs because DreamBooth uses a rare token for the new concept while keeping its textual embedding fixed, thereby leading to insufficient learning of the embedding alignment for the new concept.

**A Naive Solution.** As analyzed previously, Textual Inversion and DreamBooth exhibit distinct issues related to the embedding alignment: Textual Inversion tends to overfit the embedding alignment for the new concept, while DreamBooth demonstrates insufficient learning of the embedding alignment. A straightforward solution is to combine Textual Inversion with DreamBooth by jointly tuning the textual embedding and the U-Net, a method we denote as TI+DB. We observe that TI+DB enhances performance over Textual Inversion or DreamBooth individually. However, it still tends to neglect the learned concept when integrating it into new prompts. This issue arises from the slow update of the textual embedding relative to the U-Net. As illustrated in Figure 4, the learned textual embedding remains very close to its initial state. Furthermore, we calculate the cosine similarity between the learned and initial embeddings, which averages about 0.9997, indicating that TI+DB still suffers from insufficient learning of the embedding alignment.

### 4.2 AttnDreamBooth

To address the issues described in Section 4.1, we propose a method named AttnDreamBooth, inspired by two key observations. First, while Textual Inversion often fails to capture the subject identity and tends to overfit the embedding alignment for the new concept, it can effectively learn the embedding alignment in the very early stages of optimization. However, at these early stages, the model only learns a coarse cross-attention map for the new concept. Second, although DreamBooth fails to learn the embedding alignment, it can accurately capture the subject identity. Based on these observations, we propose to decompose the personalization process into three training stages: 1) learning the embedding alignment; 2) refining the attention map; and 3) acquiring the subject identity. An overview of our proposed AttnDreamBooth is illustrated in Figure 3.

**Learning the Embedding Alignment.** As previously stated, learning the embedding alignment for the new concept is critical for properly allocating the cross-attention maps for novel prompts, which in turn influences the text alignment of the personalized generation results. To achieve this, we optimize the input textual embedding of the text encoder, since the text encoder manages the contextual understanding of the prompt. However, as analyzed in Section 4.1, this approach is prone to overfitting the embedding, leading to an embedding misalignment issue. Therefore, our objective at this stage is to learn the embedding alignment while minimizing the risk of overfitting. To this end, we adapt Textual Inversion [24] with three main modifications. First, we significantly reduce the number of optimization steps (to 60 steps in our experiments) and lower the learning rate (to $10^{-3}$). Second, we introduce a cross-attention map regularization (see Section 4.3) to guide the learning of the cross-attention map. Third, to facilitate the incorporation of the cross-attention map regularization, we set the training prompt as "a photo of a [V] [super-category]". To prevent overfitting, we stop the optimization at very early stages, thereby resulting in a coarse cross-attention map for the new concept, as depicted in Figure 5. A full analysis of attention map allocations for each token is

| Input | Prompt | Stage 1 | Stage 2 | Stage 3 |

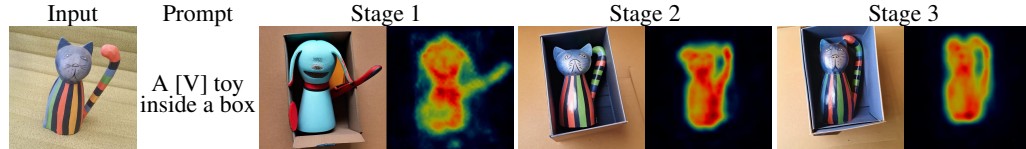

Figure 5: **Results after each training stage**. We present the generations along with the attention maps of "[V]" for each stage. In stage 1, the model properly aligns the embedding of [V] with other tokens, "inside a box", but learns a very coarse attention map and subject identity. In stage 2, the model refines the attention map and subject identity. In stage 3, the model accurately captures the identity of the concept.

presented in Appendix E. The cross-attention map, as well as the subject identity, are addressed in subsequent steps.

**Refining the Cross-Attention Map.**    To mitigate the embedding misalignment issue, our model initially learns a relatively coarse cross-attention map for the new concept. At this stage, we focus on refining the cross-attention map. Since these attention maps are embedded within the cross-attention layers, inspired by Custom Diffusion [50], we fine-tune all the cross-attention layers in the U-Net. Additionally, we employ the proposed cross-attention map regularization (see Section 4.3) to aid in refining the attention map. Furthermore, we keep the textual embedding and the text encoder fixed to prevent further embedding misalignment.

**Capturing the Subject Identity.**    As illustrated in Figure 5, the previous stage produces images that are similar to the target concept but still exhibit significant distortions. Therefore, in the third stage, following DreamBooth [71], we unfreeze all layers of the U-Net to more accurately capture the subject identity of the target concept. We choose not to adopt the prior preservation loss [71], as we empirically find that it leads to poor identity preservation and requires significantly more training steps. A detail discussion of our models with or without the prior preservation loss could be found in Appendix H. Moreover, similar to the previous stage, we keep the textual embedding and the text encoder fixed to prevent embedding misalignment, and we continue to apply the cross-attention map regularization to guide the learning of the attention map.

### 4.3 Cross-Attention Map Regularization

We set the training prompt as "a photo of a [V] [super-category]", where [V] and [super-category] denote the tokens for the new concept and its super-category, respectively. To enhance the learning of the attention map, we introduce a regularization term that encourages similarity between the attention maps of [V] and [super-category]. This regularization term serves two purposes. First, since the new concept and its super-category belong to the same object category, the attention map of the super-category token can serve as a reference for the new concept. Second, since [V] and [super-category] are used together to describe the new concept when integrating it into new prompts, the attention maps of [V] and [super-category] should refer to the same region.

Formally, for the 16 attention maps $\{M_1, M_2..., M_{16}\}$ from 16 different cross-attention layers, we minimize the squared differences in the mean and variance of the attention map values for [V] and [super-category] as follows:

$$\mathcal{L}_{\text{reg}} = \lambda_\mu \big[\mu(M_{1:16}^{\text{V}}) - \mu(M_{1:16}^{\text{category}})\big]^2 + \lambda_\sigma \big[\sigma^2(M_{1:16}^{\text{V}}) - \sigma^2(M_{1:16}^{\text{category}})\big]^2, \qquad (2)$$

where $\mu(M_{1:16})$ and $\sigma^2(M_{1:16})$ denote the mean and variance of all the values across the 16 attention maps, respectively. This constraint helps ensure that the new concept exhibits a similar level of concentration or dispersion in the attention map as the super-category token. Note that we avoid directly applying the constraint to the attention map values themselves because we empirically find that such a constraint is too restrictive and difficult to optimize.

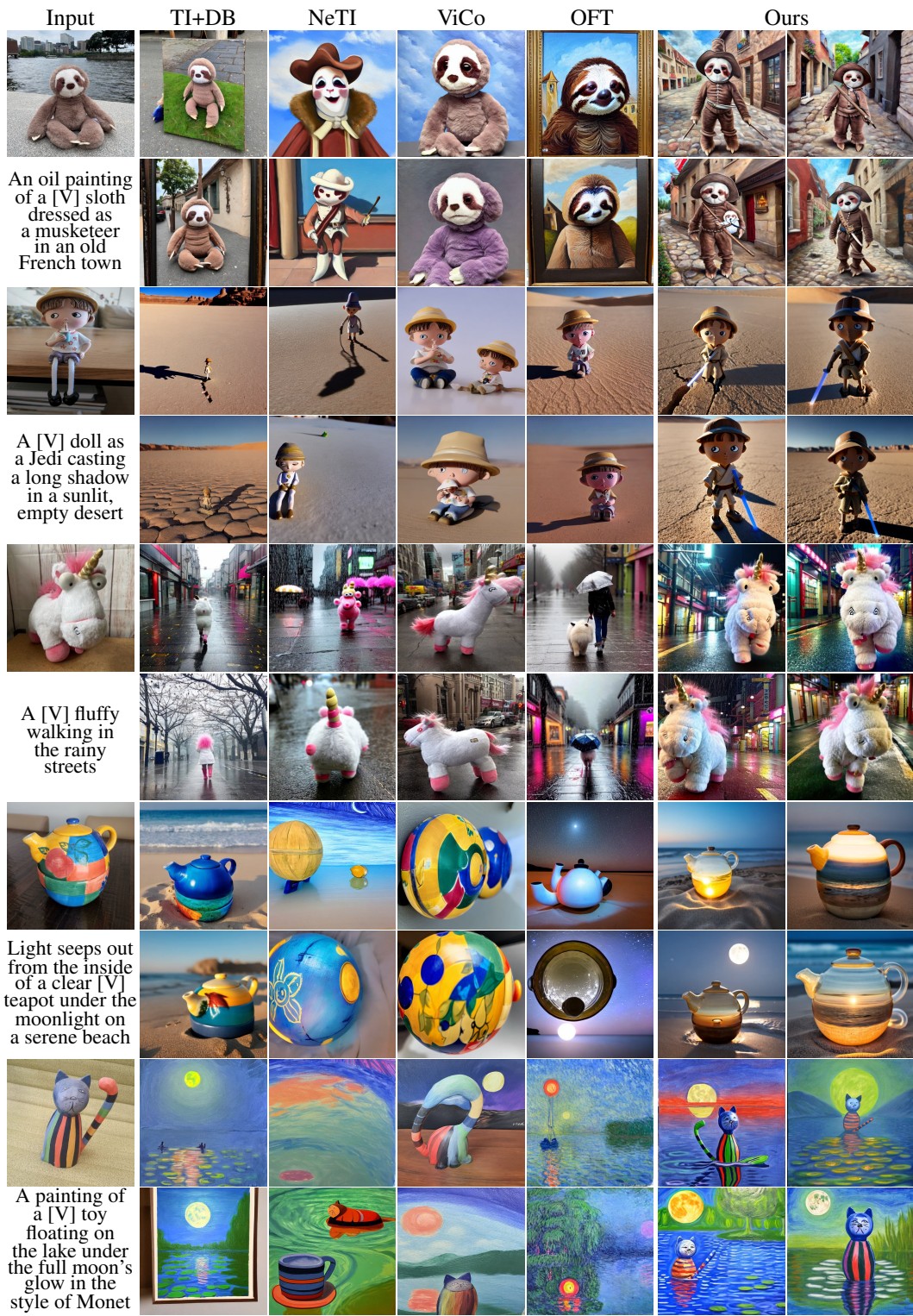

Figure 6: **Qualitative comparison**. We present four images generated by our method and two images from each of the baseline methods, including TI+DB [24, 71], NeTI [1], ViCo [30], and OFT [64]. Our method demonstrates superior performance in text alignment and identity preservation compared to these baselines.

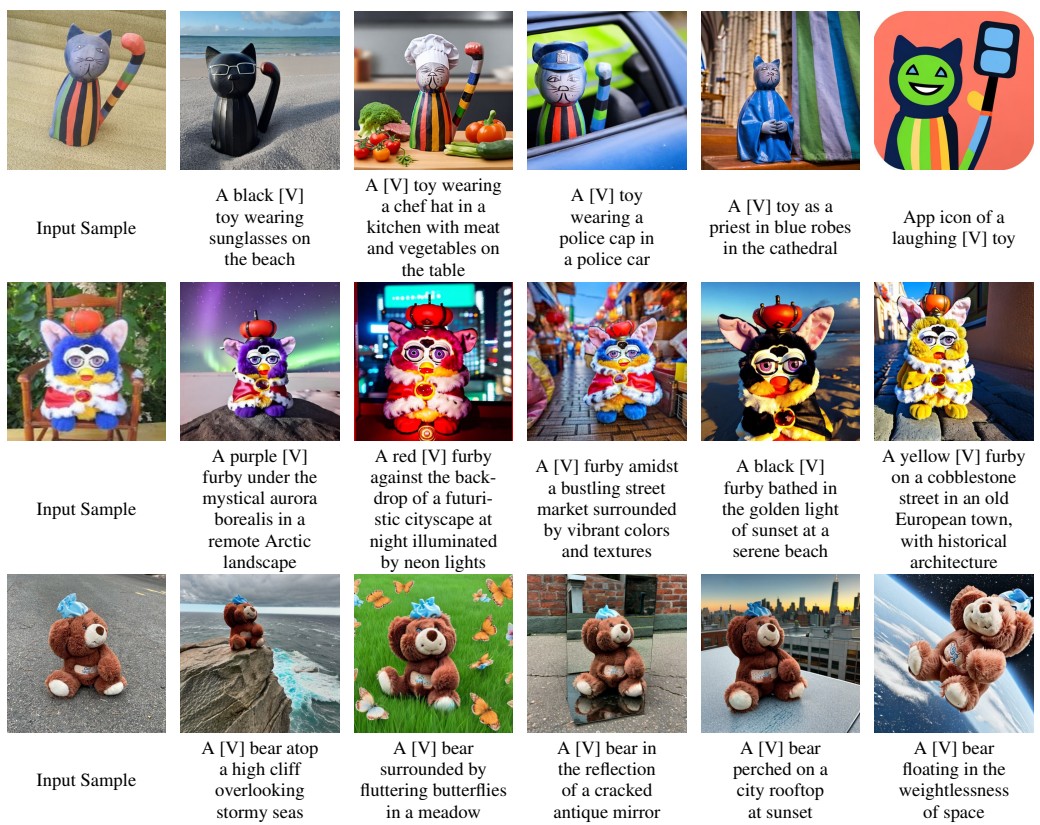

| Input Sample | A black [V] toy wearing sunglasses on the beach | A [V] toy wearing a chef hat in a kitchen with meat and vegetables on the table | A [V] toy wearing a police cap in a police car | A [V] toy as a priest in blue robes in the cathedral | App icon of a laughing [V] toy |

| Input Sample | A purple [V] furby under the mystical aurora borealis in a remote Arctic landscape | A red [V] furby against the backdrop of a futuristic cityscape at night illuminated by neon lights | A [V] furby amidst a bustling street market surrounded by vibrant colors and textures | A black [V] furby bathed in the golden light of sunset at a serene beach | A yellow [V] furby on a cobblestone street in an old European town, with historical architecture |

| Input Sample | A [V] bear atop a high cliff overlooking stormy seas | A [V] bear surrounded by fluttering butterflies in a meadow | A [V] bear in the reflection of a cracked antique mirror | A [V] bear perched on a city rooftop at sunset | A [V] bear floating in the weightlessness of space |

Figure 7: Examples of personalized generations obtained using AttnDreamBooth.

## 5 Experiments

In this section, we first present the implementation details of our method. Subsequently, we evaluate its performance by conducting a comparative analysis with four state-of-the-art personalization methods. Lastly, we conduct an ablation study to demonstrate the effectiveness of each sub-module.

### 5.1 Implementation and Evaluation Setup

**Implementation Details.** Our implementation is based on the publicly available Stable Diffusion V2.1 [70]. The textual embedding of the new concept is initialized using the embedding of the super-category token. We keep a fixed batch size of 8 across all training stages but vary the learning rates and training steps. Specifically, we train with a learning rate of $10^{-3}$ for 60 steps in stage 1, followed by a learning rate of $2 \times 10^{-5}$ for 100 steps in stage 2, and conclude with a learning rate of $2 \times 10^{-6}$ for 500 steps in stage 3. $\lambda_\mu$ and $\lambda_\sigma$ are set to 0.1 and 0 in stage 1, respectively, and are adjusted to 2 and 5 in subsequent stages. All experiments are conducted on a single Nvidia A100 GPU. The training process of our method takes about 20 minutes to learn a concept.

**Evaluation Setup.** We compare our method with four state-of-the-art personalization methods, including TI+DB [24, 71], NeTI [1], ViCo [30], and OFT [64]. The implementation details of the baseline methods are provided in Appendix A. We collect 22 concepts from TI [24] and DB [71]. For the quantitative evaluation, each method is evaluated using a set of 24 text prompts, see Appendix B for a complete list. These prompts cover background change, environment interaction, concept color change, and artistic style.

### 5.2 Results

**Qualitative Evaluation.** In Figure 6, we present a visual comparison of personalized generation for various concepts. We employ a set of complex prompts for evaluation, where one prompt simultane-

Table 1: **Quantitative comparisons**. "Identity" denotes the identity preservation, and "Text" denotes the text alignment.

| Methods | Identity↑ | Text↑ |
|---|---|---|
| TI+DB [24, 71] | 0.7017 | **0.2578** |
| NeTI [1] | 0.6901 | 0.2522 |
| ViCo [30] | **0.7507** | 0.2106 |
| OFT [64] | 0.7257 | 0.2445 |
| Ours-fast | 0.7268 | 0.2536 |
| Ours | 0.7257 | 0.2532 |

Table 2: **User study**. We asked the participants to select the image that better preserves the identity and matches the prompt.

| Baselines | Prefer Baseline | Prefer Ours |
|---|---|---|
| TI+DB [24, 71] | 32.0% | **68.0%** |
| NeTI [1] | 20.6% | **79.4%** |
| ViCo [30] | 16.6% | **83.4%** |
| OFT [64] | 22.4% | **72.6%** |

ously incorporates several editing elements such as style change (e.g., "oil painting"), scene change (e.g., "old French town"), and appearance change (e.g., "dressed as a musketeer"). As observed, ViCo tends to overfit the new concept, failing to compose it in novel scenes or styles. Conversely, TI+DB sometimes overlooks the learned concept, producing images that solely reflect other prompt tokens. NeTI and OFT also struggle to achieve text-aligned generations, especially when the prompts are complex. Our method, AttnDreamBooth, is the only method that successfully generates identity-preserved and text-aligned personalized images for these complex prompts. Figures 1 and 7 show more personalized generations using complex prompts from our method. Additional qualitative results can be found in Appendix C.

**Quantitative Evaluation.** We conduct a quantitative evaluation of each method in terms of identity preservation and text alignment. Identity preservation is measured by the cosine similarity between the CLIP [65] embeddings of generated and real images, while text alignment is measured by the cosine similarity between the CLIP embeddings of generated images and their corresponding prompts. Each method is evaluated using 24 text prompts, generating 32 images per prompt. The results are presented in Table 1. TI+DB excels in text alignment but performs poorly in identity preservation. This is consistent with the qualitative observation that TI+DB often neglects the learned concept and generates images based solely on other prompt tokens. In contrast, ViCo achieves the best identity preservation but ranks lowest in text alignment, indicative of its tendency to overfit the new concept. Besides these two extreme cases, our approach exhibits superior performance in both identity preservation and text alignment.

**Fast Version of Our Method.** The average training time using our method is about 20 minutes. To reduce the training time, we developed a fast version of our method by increasing the learning rate while simultaneously decreasing both the training steps and the batch size for the third training stage. Originally, this stage involves performing 500 steps with a learning rate of $2 \times 10^{-6}$ and a batch size of 8. The fast version now completes training in just 200 steps with a learning rate of $1 \times 10^{-5}$ and a batch size of 4. These adjustments significantly reduce the training time from 20 minutes to an average of 6 minutes. Interestingly, this fast version maintains performance comparable to our original model, likely because the first two stages provide a convenient starting point, allowing for a higher learning rate in the third stage. Notably, the fast version performs similarly to the original model on short prompts, as shown in Table 1, but it shows slight degradation on complex prompts, as depicted in Figure 10.

**User Study.** We further evaluate our method by conducting a user study. Personalized images are generated using various prompts and concepts for each method. In each question of the study, participants are presented with an input image and a text prompt, along with two generated images: one from our method and another from a baseline method. Participants are asked to select the image that better achieves identity preservation and text alignment. We collected a total of 700 responses from 35 participants, as presented in Table 2. The results demonstrate a clear preference for our method.

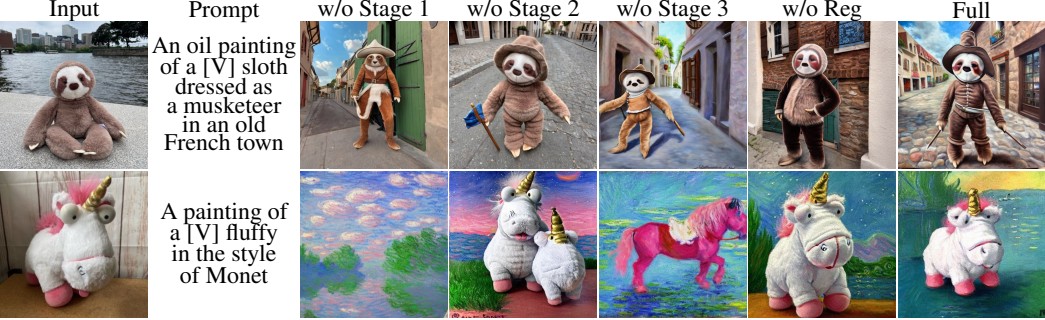

| Input | Prompt | w/o Stage 1 | w/o Stage 2 | w/o Stage 3 | w/o Reg | Full |

Figure 8: **Ablation study**. We compare models trained without optimizing the textual embedding (w/o Stage 1), without fine-tuning the cross-attention layers (w/o Stage 2), without fine-tuning the U-Net (w/o Stage 3), and without the cross-attention map regularization (w/o Reg). As can be observed, all sub-modules are essential for achieving identity-preserved and text-aligned personalized generations.

## 5.3 Ablation Study

In this section, we evaluate the effectiveness of each sub-module within our framework. Specifically, we conduct an ablation study by separately removing each training stage or the attention map regularization term. Figure 8 presents a visual comparison of personalized images generated by each variant. The results indicate that all sub-modules are crucial for achieving identity-preserved and text-aligned personalized generations. Specifically, the model without optimizing the textual embedding (w/o Stage 1) tends to neglect the learned concept or generate it with significant distortions due to insufficient learning of the embedding alignment. Models without fine-tuning the cross-attention layers (w/o Stage 2) or without the regularization term (w/o Reg) suffer from degraded text alignment or identity preservation. The model without fine-tuning the U-Net (w/o Stage 3) leads to significant degradation in identity preservation. Additional ablation study results are provided in Appendix F. Similar behavior is observed in the quantitative ablation study, as detailed in Table 4.

## 6 Conclusions and Limitations

In this paper, we identified and analyzed the embedding misalignment issue encountered by Textual Inversion and DreamBooth. Our proposed method, named AttnDreamBooth, addresses this issue by decomposing the personalization process into three stages: learning the embedding alignment, refining the attention map, and acquiring the subject identity. AttnDreamBooth enables identity-preserved and text-aligned text-to-image personalization, even with complex prompts.

In our experiments, we used consistent training steps across different concepts; however, we observed that performance could be further improved by tuning the training steps for specific concepts. This limitation might be addressed by adopting adaptive training strategies, which we leave for future work. A second limitation is that our three-stage training method requires approximately 20 minutes on average to learn a concept, as it involves fine-tuning all parameters in the U-Net for 500 steps. To mitigate this, we introduced a fast version that reduces the training time to approximately 6 minutes while still maintaining performance comparable to the original model.

## Acknowledgments

This work is supported by National Natural Science Foundation of China (No. 62176223 and No. 62302535), Guangdong Basic and Applied Basic Research Foundation (No. 2023A1515012897 and No. 2023A1515011639), and Zhuhai Basic and Applied Basic Research Foundation (No. 2320004002745).

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

## A   Implementation Details of Baselines

We compare our method with four baseline methods, including TI+DB [24, 71], NeTI [1], ViCo [30], and OFT [64]. For TI+DB, we implement it based on the diffusers library [80] without employing the prior preservation loss. We perform 660 training steps, which matches the total number of steps for our method, with a learning rate of $2 \times 10^{-6}$ and a batch size of 8. For the other baselines, we use the official implementations and follow the hyper-parameters described in their papers.

In the Appendix, we further compare our method with four other baseline methods, including DreamMatcher [58], FreeCustom [19], SuTI [10], and Instruct-Imagen [35]. For DreamMatcher and FreeCustom, we use their official implementations. Due to the unavailability of open-source models for SuTI and Instruct-Imagen, we rely on the examples provided in their papers for comparison.

## B   Text Prompts

In Table 3, we list all 24 text prompts used in the quantitative evaluation. These prompts cover a range of modifications, including background change, environment interaction, concept color change, and artistic style.

## C   Additional Qualitative Results

In Figures 9, 10, and 11, we provide additional qualitative comparisons to the baseline methods across a wide range of prompts. Furthermore, Figure 12 presents a qualitative comparison of our method with Textual Inversion (TI), DreamBooth (DB), and two different configurations of TI+DB: 1) first TI and then DB (TI→DB), and 2) first DB and then TI (DB→TI). In Figure 13, we provide additional qualitative results generated by AttnDreamBooth.

## D   Single Image Personalization

In this section, we compare AttnDreamBooth with the baseline methods when only a single image is used for training. In Figure 14, we present the generation results of each method under this challenging setting. Our method demonstrates superior text alignment and identity preservation compared to the baselines.

## E   Attention Maps for Each Token

In the main text, we provide the cross-attention maps of Textual Inversion and DreamBooth in Figure 2 and the cross-attention map of "[V]" for each stage in Figure 5. To vividly demonstrate the efficacy of our method in training the cross-attention layer, we present the generated images along with the cross-attention maps for each token in the prompt in Figure 15. As can be seen, our method accurately assigns the attention maps for each token, demonstrating correct embedding alignment for the new concept.

## F   Additional Ablation Study

In Figure 16, we provide additional ablation study results for each variant of our method. Table 4 presents the quantitative results of our ablation study. Specifically, the model without Stage 1 achieves better text alignment but significantly poorer identity preservation compared to the full model. This is because, without sufficient training of the textual embedding, the model tends to overlook the learned concept or generate it with significant distortions. Please note that the text alignment score is calculated without considering the new concept; therefore, omitting the new concept can inadvertently boost this score. Similarly, models without Stage 2 or Stage 3 also exhibit higher text alignment scores but lower identity preservation scores, due to insufficient learning of the attention maps and the subject identity, respectively. Additionally, the model without the regularization term shows degraded text alignment.

## G   Different Version of Stable Diffusion

A visual comparison between our models with SD1.5 or SD2.1 is presented in Figure 17. As shown, the model with SD2.1 achieves superior performance in text alignment and identity preservation. Nevertheless, our method is also effective for SD1.5, and it outperforms the baseline methods.

## H   Prior Preservation Loss

We present a visual comparison between models with or without the prior preservation loss in Figure 18. The results show that incorporating the prior preservation loss leads to degradation in identity preservation.

## I   User Study

As described in Section 5.2, we conducted a user study to evaluate our method against the baseline methods. Here, we present the details of this user study. Figure 19 shows an example question from the user study. Given a concept image and a text prompt, along with two generated images (one from our method and another from a baseline method), participants were asked to select the image that better preserves the identity of the concept image and aligns with the text prompt. The results are presented in Table 2.

## J   Societal Impact

Similar to existing text-to-image personalization techniques, our approach provides broader users access to effectively fine-tuning large-scale pre-trained diffusion models. By enabling users to personalize these models with their own data, our approach can be used for numerous applications, including image editing, artistic creations, and industrial production. However, the use of generative techniques comes with risks, such as the creation of misleading or false information. To mitigate these concerns, it is vital to develop effective methods for identifying fake generations [84, 15].

## K   Licenses for Pre-trained Models and Datasets

Our implementation is based on the publicly available Stable Diffusion V2.1 [70], which is under the CreativeML Open RAIL++-M License. The datasets used for evaluation are from TI [24] and DB [71]. The data from DB is under the Unsplash license, while the license information for the data from TI is not available online.

Table 3: The prompts used in the quantitative evaluation.

| a photo of a [V] [category] |
|---|
| a photo of a [V] [category] in Times Square |
| a photo of two [V] [category] on a table |
| a [V] [category] in the jungle |
| a [V] [category] on a stone wall in the countryside |
| a [V] [category] on a brick pathway in a garden |
| a [V] [category] on a pile of fallen leaves in a forest |
| a [V] [category] at a picnic spot with a checkered blanket |
| a [V] [category] nestled among rocks |
| a [V] [category] inside a basket |
| a [V] [category] inside a metal cage |
| a [V] [category] drenched in the rainy streets |
| a [V] [category] in a grassy park with a sunglasses |
| a [V] [category] floats on the water |
| a [V] [category] covered by snow |
| a red [V] [category] wearing bowtie |
| a purple [V] [category] |
| a black [V] [category] |
| a [V] [category] latte art |
| pencil drawing of a [V] [category] |
| manga drawing of a [V] [category] |
| a watercolor painting of a [V] [category] |
| vector art of a [V] [category] |
| a painting of a [V] [category] in the style of Monet |

Table 4: Quantitative ablation study.

| Methods | Identity Preservation↑ | Text Alignment↑ |
|---|---|---|
| W/o Stage 1 | 0.7031 | 0.2595 |
| W/o Stage 2 | 0.7145 | 0.2541 |
| W/o Stage 3 | 0.6821 | 0.2650 |
| W/o Reg | 0.7269 | 0.2502 |
| Full Model | 0.7257 | 0.2532 |

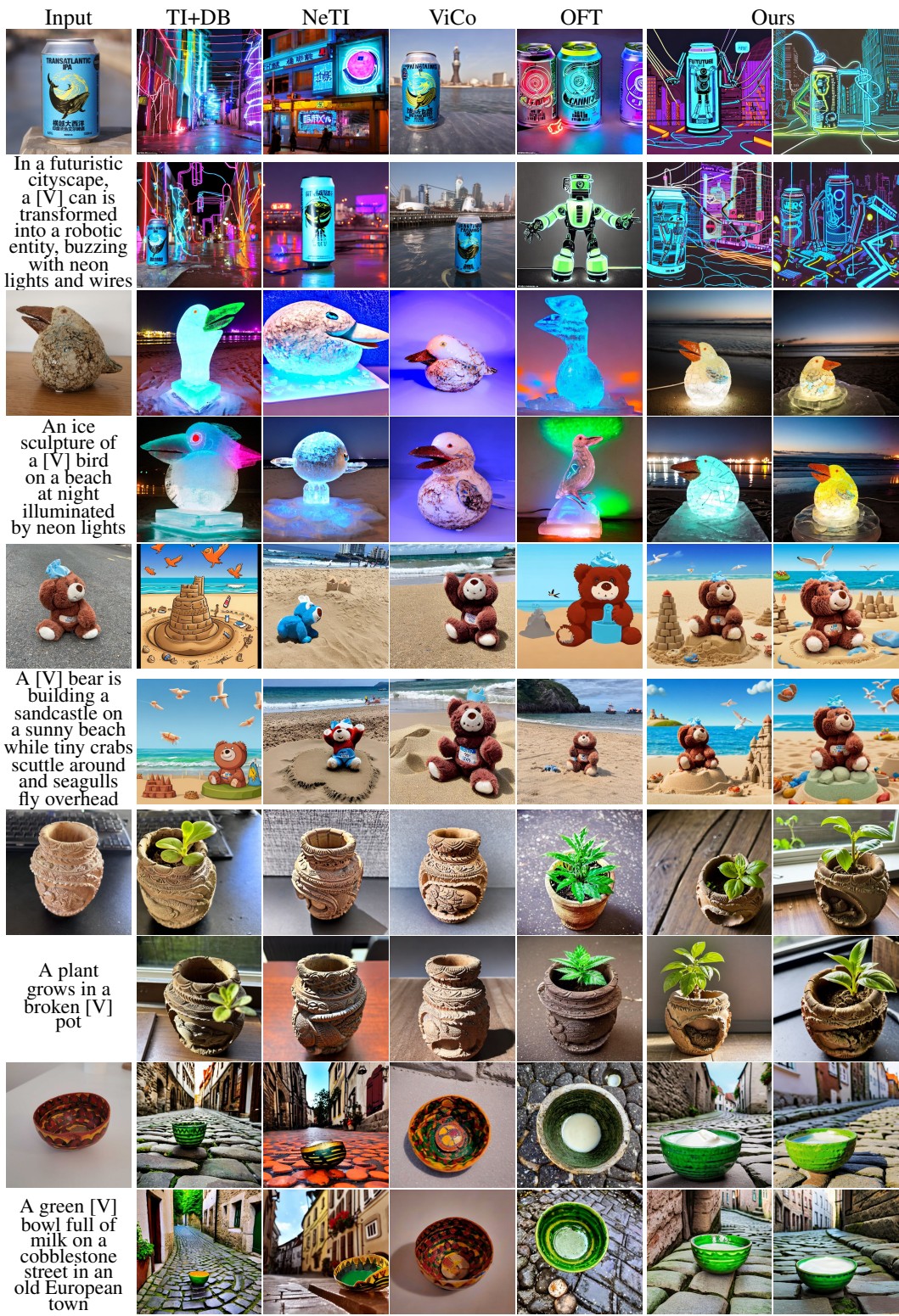

Figure 9: **Additional qualitative comparison**. We present four images generated by our method and two images from each of the baseline methods, including TI+DB [24, 71], NeTI [1], ViCo [30], and OFT [64]. Our method demonstrates superior performance in text alignment and identity preservation compared to these baselines.

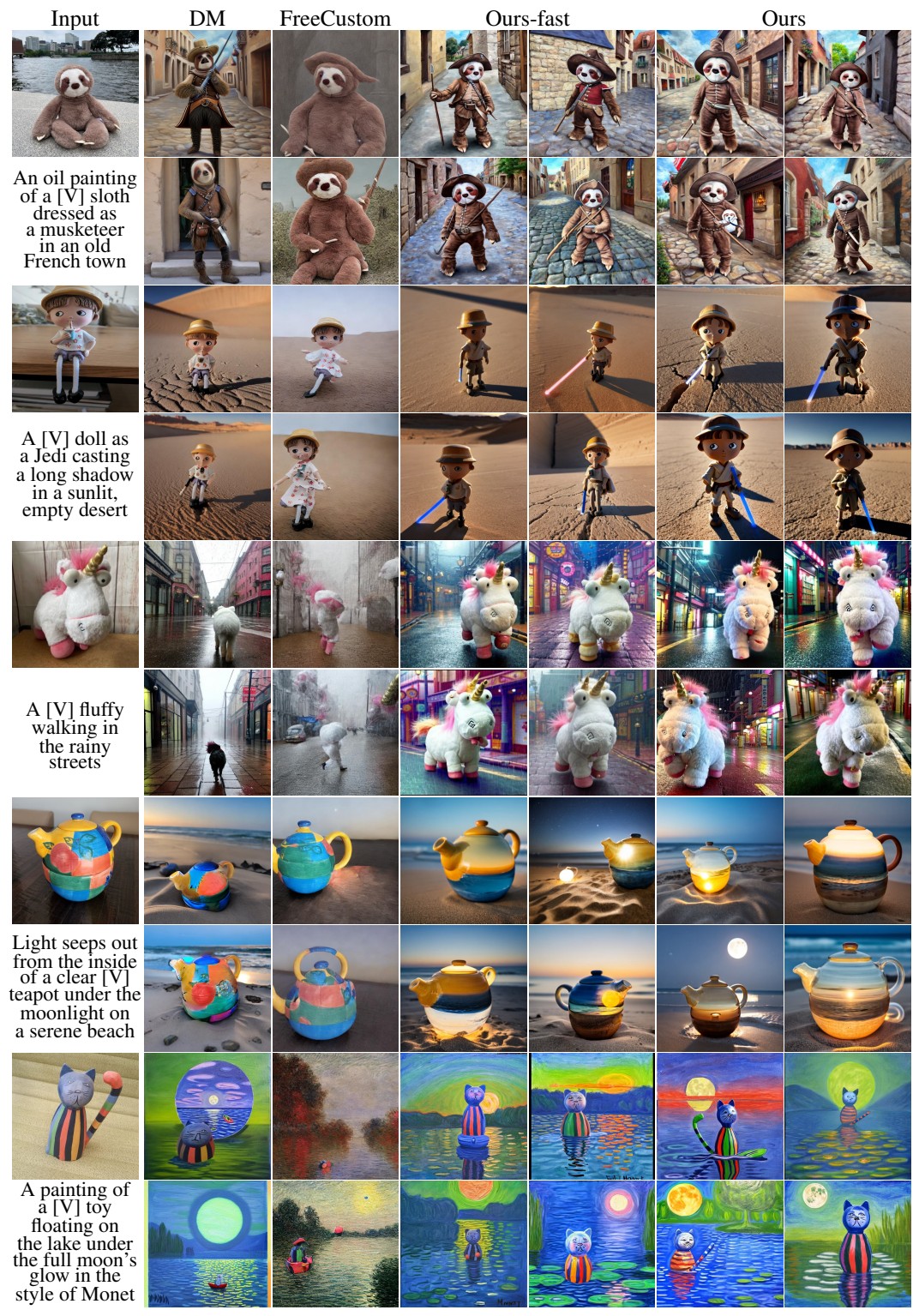

Figure 10: Qualitative results of the fast version of our method, compared with DreamMatcher (DM) [58] and FreeCustom [19].

| Input | SuTI | Ours | Input | SuTI | Ours |
|-------|------|------|-------|------|------|

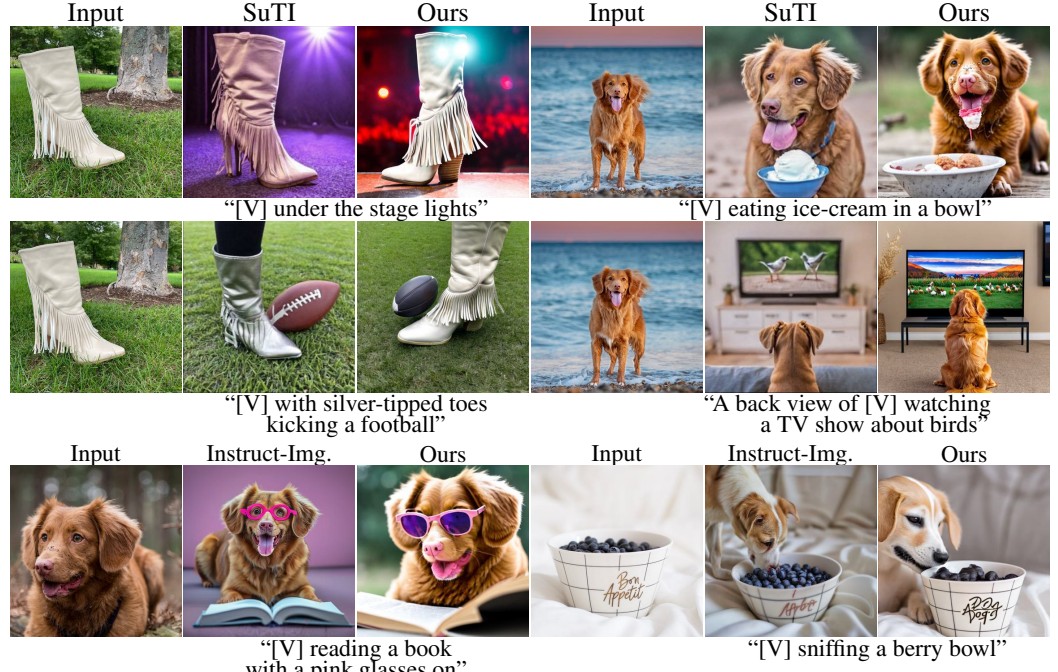

"[V] under the stage lights"      "[V] eating ice-cream in a bowl"

"[V] with silver-tipped toes kicking a football"      "A back view of [V] watching a TV show about birds"

"[V] reading a book with a pink glasses on"      "[V] sniffing a berry bowl"

Figure 11: Qualitative comparison to SuTI [10] and Instruct-Imagen [35] using the examples from their papers.

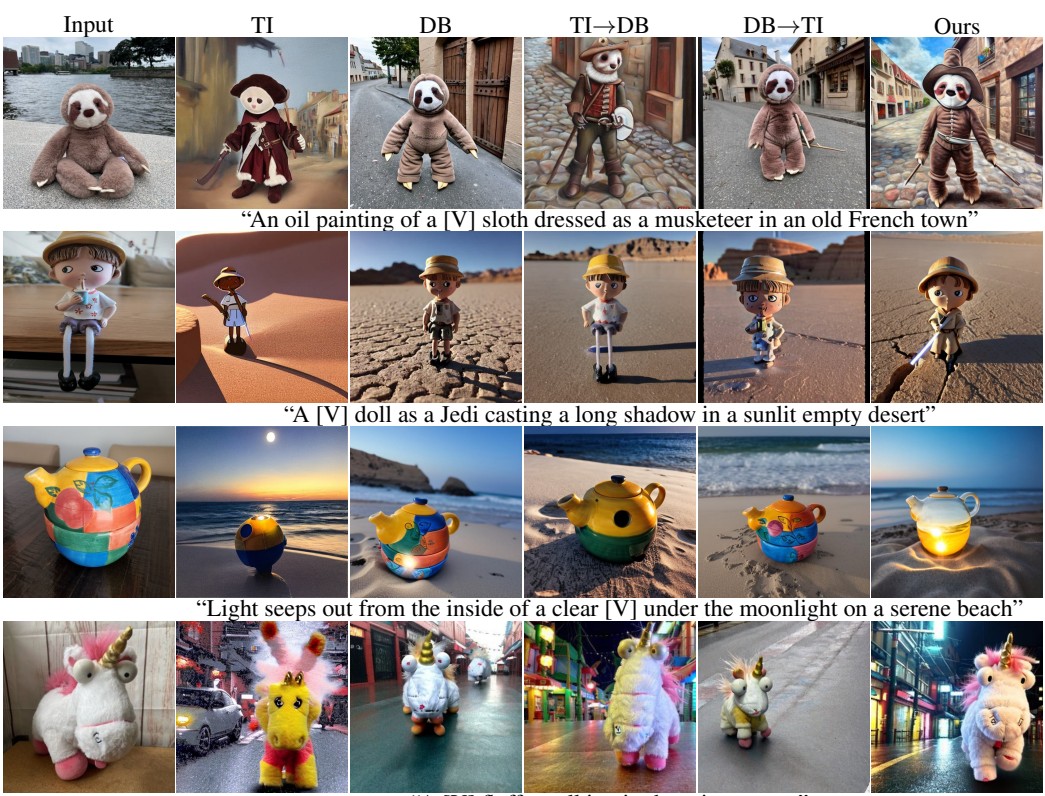

"An oil painting of a [V] sloth dressed as a musketeer in an old French town"

"A [V] doll as a Jedi casting a long shadow in a sunlit empty desert"

"Light seeps out from the inside of a clear [V] under the moonlight on a serene beach"

"A [V] fluffy walking in the rainy streets"

Figure 12: Qualitative comparison of our method with Textual Inversion (TI), DreamBooth (DB), and two different configurations of TI+DB: 1) first TI and then DB (TI→DB), and 2) first DB and then TI (DB→TI).

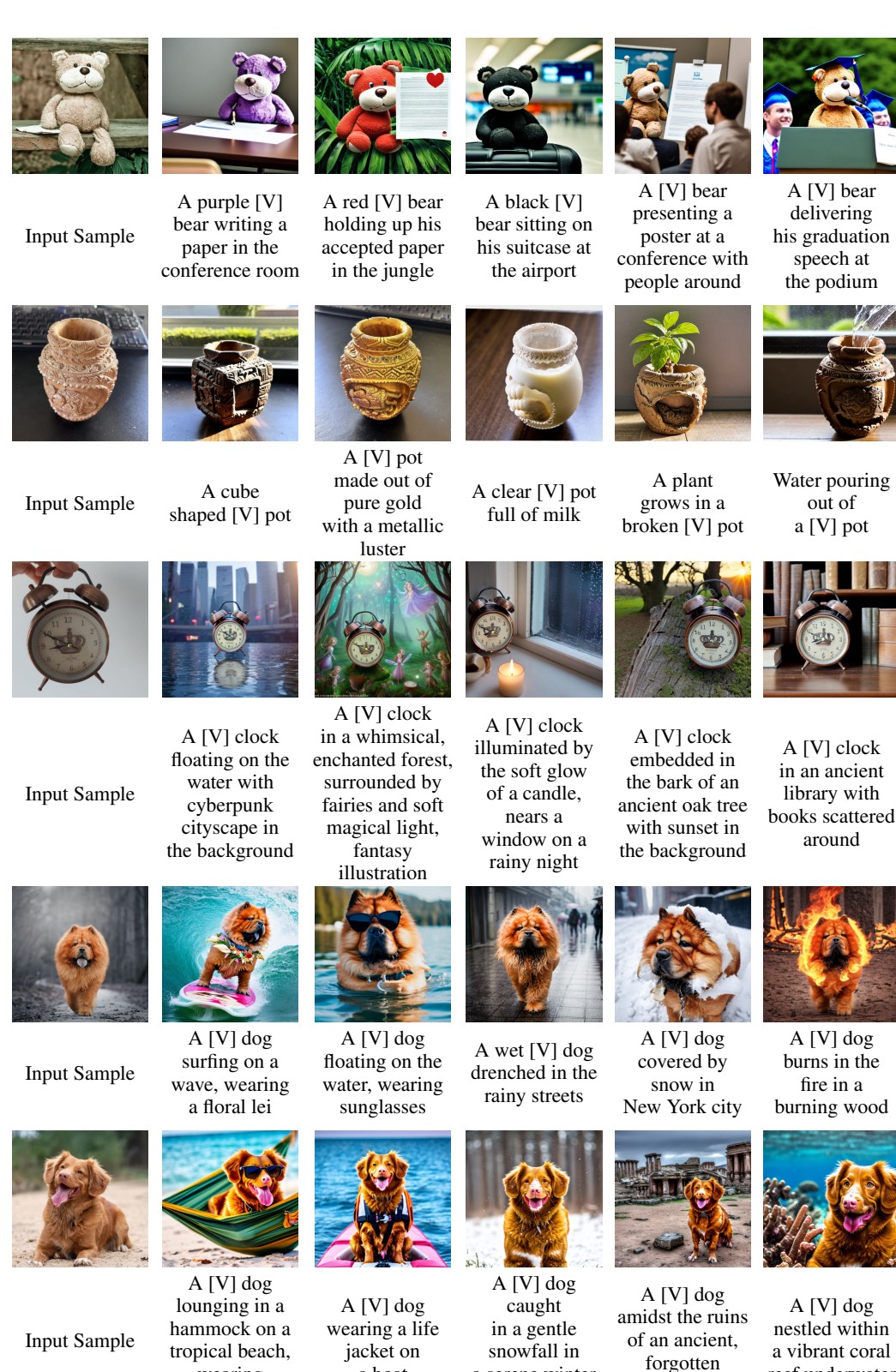

Figure 13: Additional qualitative results by AttnDreamBooth.

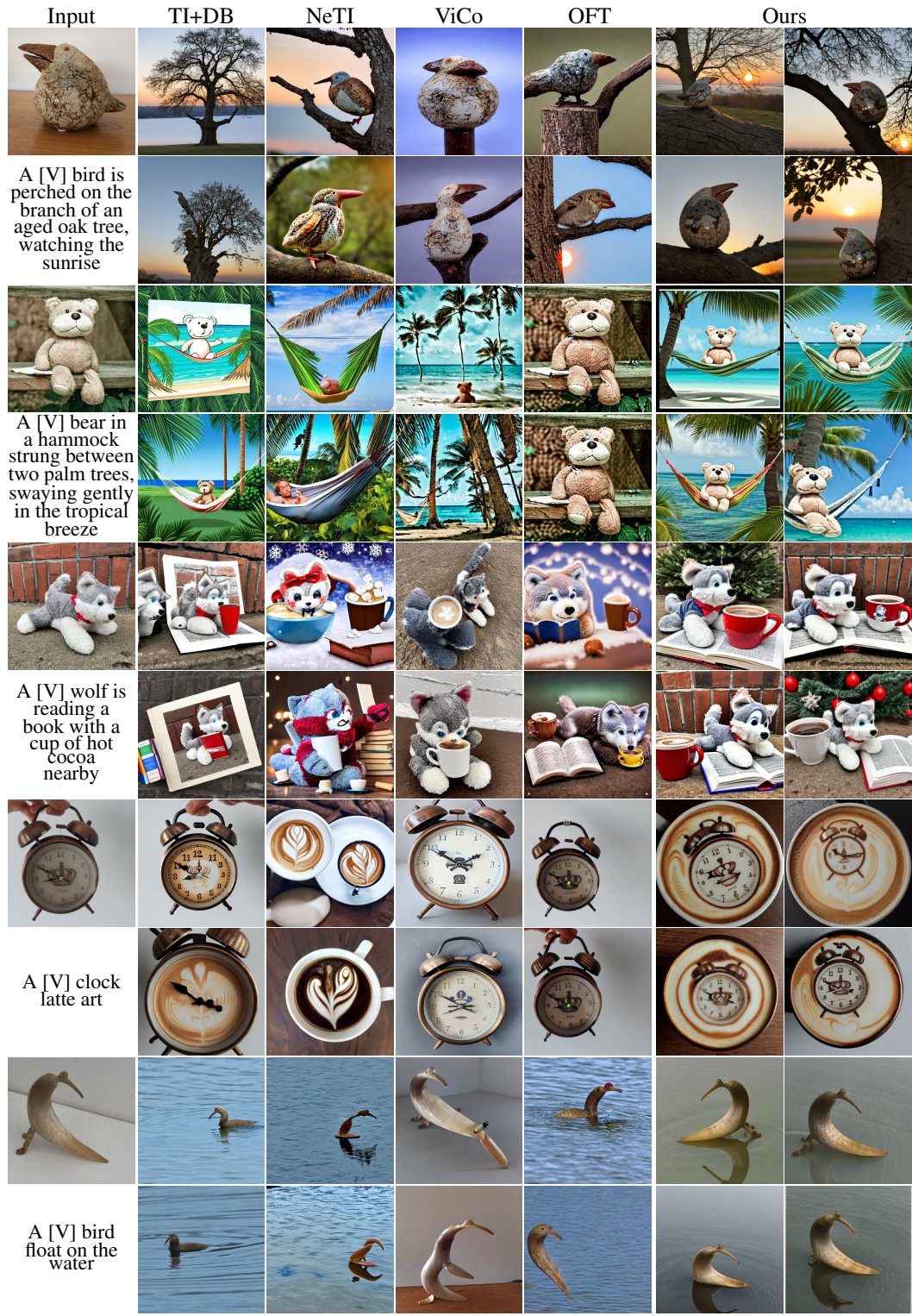

Figure 14: **Single image personalization results.** We present four images generated by our method and two images from each of the baseline methods, including TI+DB [24, 71], NeTI [1], ViCo [30], and OFT [64]. Our method shows better text alignment and identity preservation than the baselines.

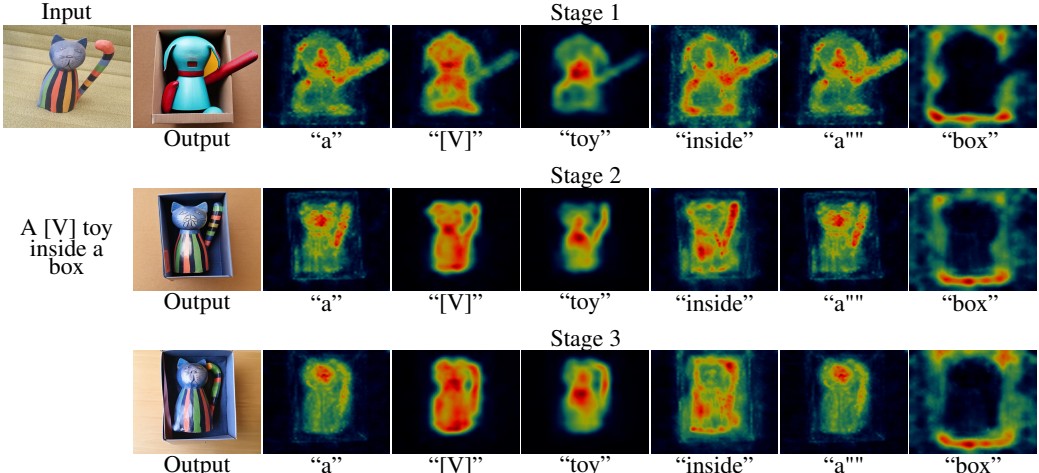

Figure 15: The attention maps for each token after each training stage.

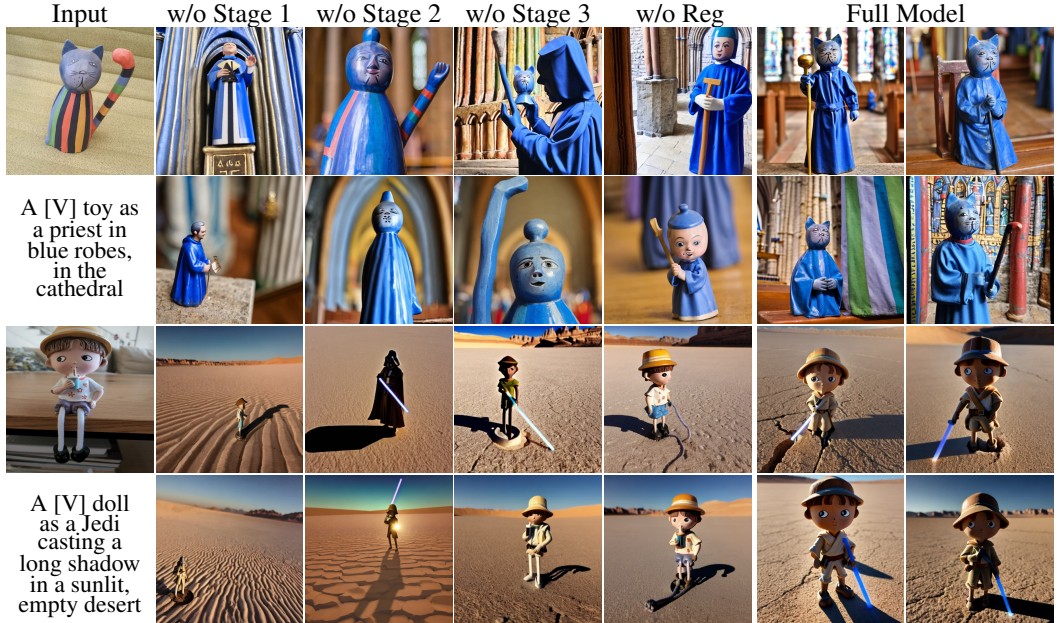

Figure 16: Additional ablation study results.

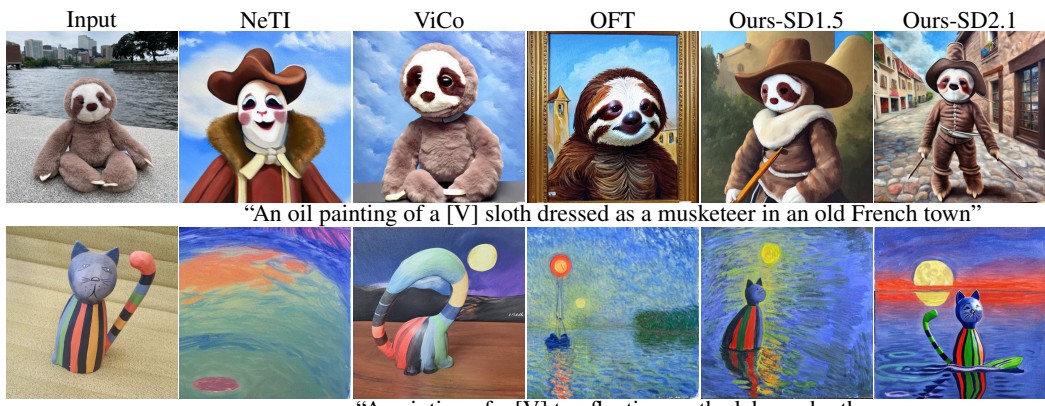

"An oil painting of a [V] sloth dressed as a musketeer in an old French town"

"A painting of a [V] toy floating on the lake under the full moon's glow in the style of Monet"

Figure 17: Results of our model using SD1.5.

Input      With prior preservation loss      Without prior preservation loss

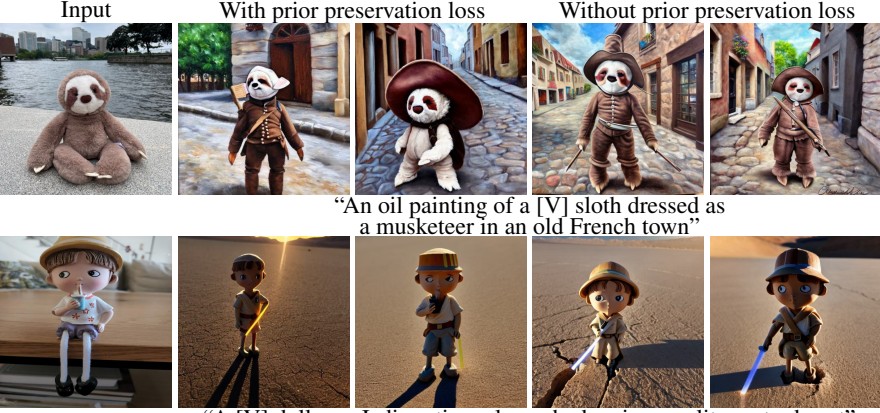

"An oil painting of a [V] sloth dressed as
a musketeer in an old French town"

"A [V] doll as a Jedi casting a long shadow in a sunlit empty desert"

Figure 18: Results of our models with or without the prior preservation loss.

\* 1. Concept image (denoted as [V]):

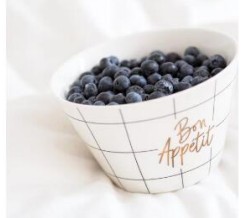

Text prompt: "A [V] burns in the fire".

Please select the image from the options below that better preserves the identity of the
concept image shown above and aligns with the text prompt provided.

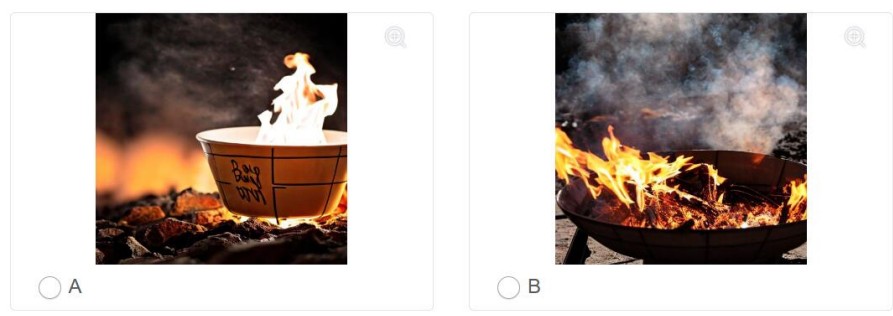

○ A          ○ B

Figure 19: **An example question of the user study**. Given a concept image and a text prompt, along
with two generated images, participants are asked to select the image that better preserves the identity
of the concept image and aligns with the text prompt.

