# OpenReview forum: "AttnDreamBooth: Towards Text-Aligned Personalized Text-to-Image Generation"
_NeurIPS.cc/2024/Conference — NeurIPS 2024 poster_

### Official Review · Reviewer_FYJx · 2024-07-11

**Soundness:** 3
**Presentation:** 3
**Contribution:** 3
**Rating:** 5
**Confidence:** 4

**Summary:**

This paper proposes a new method for personalized image generation, decompose the personalization process into three training stages and introducing a cross-attention map regularization term.

**Strengths:**

The manuscript is well-written.

The authors propose to address the intrinsic issues of two classical and up-to-date methods, Textual Inversion and DreamBooth, by aligning the new concept from the prompt while preserving the original subject identity. This approach tackles a significant problem.

**Weaknesses:**

1.	Some parts of the manuscript are slightly verbose. For instance, the introduction section already introduces the existing problems of Textual Inversion and DreamBooth, along with their corresponding analysis and naïve solution. However, these points are reiterated in Section 4.1 and the first paragraph of Section 4.2 without adding any new information.

2.	What does the term 'this approach' refer to in Line 174?

3.	In general, the proposed method tends to be a bag of tricks with some customized hype-parameters.

4.	Important comparisons are lacking in the manuscript, specifically the comparison with Textual Inversion (TI) and DreamBooth (DB) individually, in addition to the comparison with the combined approach of TI+DB.

5.	This method uses cross-attention map for regularization, which results in a high time cost; on a NVIDIA A100, training requires 660 steps, taking 20 minutes. In contrast, DreamBooth requires only 5 minutes to train for 1000 steps.

**Questions:**

1.	Do the constraints imposed between [V] and [super-category] across all layers in Eq. 2 potentially restrict the diversity? Considering that [super-category] represents the original identity information and [V] introduces diversity.

2.	Why was SD2.1 chosen, and how do other models perform?

3.	In line 197, it is mentioned that retaining the prior preservation loss leads to poor results. Can you provide experimental results to support this?

---

> ### Author Rebuttal · Authors · 2024-08-07
>
> We would like to thank the reviewer for the constructive comments. We response the reviewer’s concerns as follows.
>
> >**W1: The problems and analysis of Textual Inversion and DreamBooth introduced in the Introduction section are reiterated in Section 4.1**
>
> Thanks for this point. The current version of Section 4.1 provides a more detailed problem analysis of existing methods than the Introduction section. We agree that minimizing repetition can enhance the clarity and conciseness of the paper. To improve conciseness, we will revise Sections 1 and 4.1 as follows: 1) remove the description of the naive solution from Section 1; 2) condense the analysis of existing methods in Section 1; 3) streamline the problem description in Section 4.1; and 4) enhance Section 4.1 with a more detailed analysis and deeper insights.
>
> >**W2: What does the term 'this approach' refer to in Line 174?**
>
> In Line 174, "this approach" refers to Textual Inversion, which involves optimizing the input textual embedding of the text encoder to learn the new concept. Thank you for this point, and we will revise Lines 173-174 as: "To achieve this, we choose to optimize the input textual embedding of the text encoder, as Textual Inversion does, given that the text encoder manages the contextual understanding of the prompt. However, as analyzed in Section 4.1, this approach is prone to overfitting the textual embedding, resulting in an embedding misalignment issue."
>
> >**W3: The proposed method tends to be a combination of existing methods.**
>
> We would like to clarify the key contributions of the paper. First, we identify a crucial insight in text-to-image personalization: the textual embedding of the new concept should be aligned with and seamlessly integrated into existing tokens, unlike existing studies that mainly focus on learning the new concept itself. Second, based on this insight, we propose a three-stage personalization process that is more than just a combination of existing techniques. The first stage is designed to learn the embedding alignment while mitigating the risk of overfitting, thus we significantly reduce the optimization steps to just 60. This results in a very coarse attention map and identity for the new concept. Subsequently, we refine the attention map by fine-tuning the cross-attention layers in the second stage, followed by fine-tuning the U-Net to capture the concept's identity in the third stage. Third, we propose an attention map regularization that utilizes the super-category token to guide the attention map learning in a self-supervised manner. Thanks for pointing this out, and we will improve the statements about the contributions in the revised paper.
>
> >**W4: Comparisons to Textual Inversion (TI) and DreamBooth (DB) individually.**
>
> Thanks for the suggestion. Comparisons to TI and DB are provided in Figure R5 of the attached PDF. As shown, our model achieves superior performance in both identity preservation and text alignment. We will add this comparison to the revised paper.
>
> >**W5: The proposed attention map regularization results in a high time cost. The method takes 20 minutes for 660 steps, while DreamBooth takes only 5 minutes for 1000 steps.**
>
> Indeed, computational overhead is a limitation of our method. We would like to clarify that the attention map regularization is not the most time-comsuming aspect, adding an average of 67 seconds. Instead, the third training stage consumes the most time. To mitigate this, we developed an effective strategy to reduce the training time. Due to the limited text length, please refer to our global response for the details. Our fast version model significantly reduces the training time from 20 minutes to 6 minutes, while maintaining performance comparable to the original model. We thank FYJx for this point, and will definitely include the results of this fast version model in the revised paper!
>
> >**Q1: Do the constraints imposed between [V] and [super-category] in Eq. 2 potentially restrict the diversity? Considering that [super-category] represents the original identity information and [V] introduces diversity.**
>
> We assume that the "diversity" refers to the distinctive characteristics of the new concept [V], but please correct us if we are wrong. Our proposed constraint does not restrict the diversity of [V], as it does not enforce a strong constraint between [V] and [super-category]. Instead, we apply a flexible constraint that enforces similarity in the mean and variance of the attention map values. This strategy aims to encourage [V] to exhibit a level of concentration or dispersion in the attention map similar to that of [super-category].
>
> >**Q2: Why was SD2.1 chosen, and how do other models perform?**
>
> We empirically find that SD2.1 achieves better performance than SD1.5 for our method. SD2.1 is widely used in text-to-image personalization models, such as AnyDoor [1], ADI [2], and IDAdapter [3]. A visual comparison between our models with SD1.5 or SD2.1 is presented in Figure R6 of the attached PDF. As shown, the model with SD2.1 achieves superior performance in text alignment and identity preservation. Nevertheless, our method is also effective for SD1.5, and it outperforms the baseline methods. We will add this discussion to the revised paper.
>
> >**Q3: It is mentioned that retaining the prior preservation loss leads to poor results. Can you provide experimental results to support this?**
>
> Thanks for pointing this out. We present a visual comparison between models with or without the prior preservation loss in Figure R7 of the attached PDF. The results show that incorporating the prior preservation loss leads to degradation in identity preservation. We will add this comparison to the revised paper.
>
> [1] AnyDoor: Zero-shot Object-level Image Customization, 2023
>
> [2] Learning Disentangled Identifiers for Action-Customized Text-to-Image Generation, 2023
>
> [3] IDAdapter: Learning Mixed Features for Tuning-Free Personalization of T2I Models, 2024

---

### Official Review · Reviewer_U1BU · 2024-07-13

**Soundness:** 3
**Presentation:** 3
**Contribution:** 3
**Rating:** 6
**Confidence:** 4

**Summary:**

This paper proposes a method to enhance the performance of personalizing text-to-image models by appropriately combining textual inversion approach, which learns new embeddings, and DreamBooth approach, which fine-tunes model weights. They demonstrate the effectiveness of their approach through qualitative evaluation and human evaluation.

**Strengths:**

* Research on appropriately combining textual inversion and DreamBooth methods has been needed in this field, and they propose a reasonable and clear method for this.
* The experimental results are good.

**Weaknesses:**

* (minor) The proposed method seems quite similar to the Magicapture [1] approach in that it separates embedding learning and weight fine-tuning and conducts regularization on attention. However, this paper does not cover a comparison with Magicapture.
* (minor) The ablation study is not conducted quantitatively and relies on a single generation scenario.

[1] Hyung, Junha, Jaeyo Shin, and Jaegul Choo. "Magicapture: High-resolution multi-concept portrait customization." Proceedings of the AAAI Conference on Artificial Intelligence. Vol. 38. No. 3. 2024.

**Questions:**

When conducting analysis in Figure 1, how many steps was DreamBooth trained for to obtain the results? It seems that with sufficient training steps, DreamBooth might exhibit different results.

**Limitations:**

Yes.

---

> ### Author Rebuttal · Authors · 2024-08-07
>
> We would like to thank the reviewer for the constructive comments. We response the reviewer’s concerns as follows.
>
> >**W1: (minor) The proposed method seems quite similar to the Magicapture [1] approach in that it separates embedding learning and weight fine-tuning and conducts regularization on attention. However, this paper does not cover a comparison with Magicapture.**
>
> Thanks for the suggestion to compare our method with MagiCapture [1]. Indeed, MagiCapture also employs a multi-stage learning strategy that first optimizes the textual embedding and then jointly finetunes the textual embedding and U-Net. However, our method differs in several key aspects. First, the motivation for our first stage (i.e., optimizing the textual embedding) differs from MagiCapture. We focus on learning the embedding alignment while mitigating the risk of overfitting, thus significantly reducing the optimization steps to just 60. In contrast, MagiCapture optimizes the textual embedding for 1200 steps in the first stage. Second, the attention regularization in MagiCapture is applied with the help of a user-provided mask, while we utilize the super-category token to guide the attention map learning in a self-supervised manner. Third, our learning process is divided into three stages: learning the embedding alignment, refining the attention map, and capturing the subject identity. Additionally, as MagiCapture is designed for integrating subject and style concepts, and our method focuses solely on personalizing subject concepts, a direct comparison of performance between these two methods is not feasible. We will include this comparison with MagiCapture in the revised paper.
>
>
> >**W2: (minor) The ablation study is not conducted quantitatively and relies on a single generation scenario.**
>
> Thank you for pointing this out. The table below presents the quantitative results of our ablation study. Specifically, the model without Stage 1 achieves better text alignment but significantly poorer identity preservation compared to the full model. This is because, without sufficient training of the textual embedding, the model tends to overlook the learned concept or generate it with significant distortions. Please note that the text alignment score is calculated without considering the new concept; therefore, omitting the new concept can inadvertently boost this score. Similarly, models without Stage 2 or Stage 3 also exhibit higher text alignment scores but lower identity preservation scores, due to insufficient learning of the attention maps and the subject identity, respectively. Additionally, the model without the regularization term shows degraded text alignment. Regarding qualitative evaluation, more generated images are provided in Figure 13 of the Appendix. We will include this quantitative ablation study and more generated images in the revised paper.
>
> | Methods              | Identity Preservation | Text Alignment |
> | :---------------- | :------: | ----: |
> | w/o Stage 1    |   0.7031   | 0.2595 |
> | w/o Stage 2    |   0.7145   | 0.2541 |
> | w/o Stage 3    |  0.6821   | 0.2650 |
> | w/o Reg          |  0.7269   | 0.2502 |
> | Full Model |  0.7257   | 0.2532 |
>
> >**Q1: When conducting analysis in Figure 1, how many steps was DreamBooth trained for to obtain the results? It seems that with sufficient training steps, DreamBooth might exhibit different results.**
>
> In Figure 1, we follow NeTI [2] to perform 500 steps with a batch size of 4 for training DreamBooth. The results of training DreamBooth for more steps are provided in Figure R3 of the attached PDF. Indeed, for certain examples (e.g., "Manga drawing of a [V] can"), DreamBooth can generate text-aligned images after 1,000 training steps. However, for many other examples (as shown in Figure R3), DreamBooth still tends to overlook the new concept even after 5,000 training steps. In contrast, our method successfully generates text-aligned images for these prompts. We appreciate this point and will include this discussion in the revised paper.
>
> **References**
>
> [1] Junha Hyung et. al. "Magicapture: High-resolution multi-concept portrait customization". AAAI, 2024.
>
> [2] Yuval Alaluf et. al. "A Neural Space-Time Representation for Text-to-Image Personalization". SIGGRAPH Asia, 2023.

---

### Official Review · Reviewer_hyCc · 2024-07-13

**Soundness:** 3
**Presentation:** 3
**Contribution:** 2
**Rating:** 5
**Confidence:** 4

**Summary:**

The author proposed a method to generate high-quality personalized images. First, a textual embedding is learned, then the cross-attention layers is  finetund to refine attention map during learning the  textual embedding, finally, the entire U-Net is trained to capture the subject identity.

**Strengths:**

1. The paper is welll-written and easy to understand.
2. The proposed method achieve competitive results comparing with SOTAs.
3. Using the super-category attention map is an interesting idea to calibrate the [V] attention map in a self-supervised manner.

**Weaknesses:**

1. The proposed method needs a costly test-time optimization to generate personalized images. The 3 stage finetuning requires expensive computation (20 minutes in A100)
2. Even though the Attention Map Regularization idea is interesting, it seems to be the only critical contribution in this paper. The authors combine some techniques from off-the-shelf method including CostumDiffusion/DB/TI, and conducted them step by step with proposed -attention map regularization.

**Questions:**

1. Where does the super-category label come from? Is it a part of the annotation?
2. I am curious how the  TI+DB perform if it is optimized step by step like AttnDreamBooth, e.g. first tuning the textual embedding and then the U-Net or first tuning U-Net then optimizing the embedding.

---

> ### Author Rebuttal · Authors · 2024-08-07
>
> We would like to thank the reviewer for the constructive comments. We response the reviewer’s concerns as follows.
>
> >**W1: The proposed method needs a costly test-time optimization to generate personalized images. The 3 stage finetuning requires expensive computation (20 minutes in A100).**
>
> Indeed, computational overhead is a limitation of our method. To address this, we explored a simple yet effective strategy to reduce the training time. This involves increasing the learning rate while simultaneously decreasing both the training steps and the batch size for our third training stage, which is notably the most time-consuming phase. Specifically, the third stage of our original model performs 500 steps with a learning rate of 2e-6 and a batch size of 8. The fast version now completes training in just 200 steps with a learning rate of 1e-5 and a batch size of 4. This adjustment significantly reduces the training time from 20 minutes to 6 minutes on average. Interestingly, this fast version maintains performance comparable to our original model, likely because the first two stages provide a convenient starting point, allowing for a higher learning rate in the third stage. The qualitative evaluation is provided in Figure R4 of the attached PDF, and quantitative results are detailed in the table below. We observed that the fast version model performs very closely to the original model for short prompts (e.g., it even slightly outperforms the original model in the quantitative evaluation), but it slightly underperforms for complex prompts (e.g., the second and fourth examples in Figure R4).
>
> | Methods              | Identity Preservation | Text Alignment | Training Time |
> | :---------------- | :------: | :----: | :----: |
> | NeTI [1]  		| 0.6901  | 0.2522 | 13 minutes |
> | Ours-fast       |  0.7268   | 0.2536 | 6 minutes |
> | Ours-original |  0.7257   | 0.2532 | 20 minutes |
>
> We thank hyCc for this point and will definitely include these results of this fast version model in the revised paper!
>
> >**W2: Even though the Attention Map Regularization idea is interesting, it seems to be the only critical contribution in this paper. The authors combine some techniques from off-the-shelf method including CostumDiffusion/DB/TI, and conducted them step by step with proposed attention map regularization.**
>
> We appreciate your recognition of our attention map regularization's contribution. In addition to this regularization, we would like to clarify other key contributions of our paper. First, we identify a crucial insight in text-to-image personalization: the textual embedding of the new concept should be aligned with and seamlessly integrated into existing tokens, unlike existing studies that mainly focus on learning the new concept itself. Second, based on this insight, we propose a three-stage personalization process that is more than just a combination of existing techniques. The first stage is designed to learn the embedding alignment while mitigating the risk of overfitting, thus reducing the optimization steps to just 60. This approach results in a very coarse attention map and identity for the new concept. Subsequently, we refine the attention map by fine-tuning the cross-attention layers in the second stage, followed by fine-tuning the U-Net to capture the concept's identity in the third stage.
>
> >**Q1: Where does the super-category label come from? Is it a part of the annotation?**
>
> Yes, the dataset provides a coarse descriptor for each concept. In fact, many approaches, such as Textual Inversion, DreamBooth, and NeTI, also require a super-category label to initialize the textual embedding of the new concept or to provide prior knowledge. Regarding the use of the super-category label in our attention map regularization, this method does not necessitate a precise super-category label, as it does not enforce a strong constraint between the new concept and the super-category token. Instead, we impose a flexible constraint that enforces similarity in the mean and variance of the attention map values, as illustrated in Eq. (2) of the main paper. This strategy aims to encourage the new concept to exhibit a level of concentration or dispersion in the attention map similar to that of the super-category token.
>
> >**Q2: I am curious how the TI+DB perform if it is optimized step by step like AttnDreamBooth, e.g. first tuning the textual embedding and then the U-Net or first tuning U-Net then optimizing the embedding.**
>
> In Figure R5 of the attached PDF, we present the results of the two suggested settings for TI+DB: 1) first tuning the textual embedding and then the U-Net (denoted as TI -> DB), and 2) first tuning the U-Net and then the textual embedding (denoted as DB -> TI). As shown, both models fail to generate text-aligned images. While the TI -> DB setting improves performance compared to using TI or DB individually, it still suffers from overfitting issues. The DB -> TI setting performs very closely to the DB model alone. In contrast, our method successfully generates images that preserve concept identity and align with the text.
>
> **References**
>
> [1] Yuval Alaluf et. al. "A Neural Space-Time Representation for Text-to-Image Personalization". SIGGRAPH Asia, 2023.

---

### Official Review · Reviewer_ZpWY · 2024-08-03

**Soundness:** 3
**Presentation:** 3
**Contribution:** 2
**Rating:** 3
**Confidence:** 5

**Summary:**

The submission proposes AttnDreamBooth for text-to-image personalization. It addresses the limitations of existing methods, Textual Inversion and DreamBooth, by separating the learning process into three stages: embedding alignment, attention map refinement, and subject identity capture. The method aims to improve identity preservation and text alignment in generated images.

# Key Contributions:
- Introduces an approach for text-to-image personalization.
- Addresses the limitations of existing methods by separating the learning process.
- Demonstrates improved performance in terms of identity preservation and text alignment.
- Provides a comprehensive analysis of the proposed method through qualitative and quantitative evaluations.

**Strengths:**

-The paper identifies a key challenge in text-to-image personalization and provides a novel solution.
- The proposed method, AttnDreamBooth, demonstrates superior performance in both identity preservation and text alignment compared to Textual Inversion and Dreambooth.
- The work contributes to advancing the field of text-to-image personalization by offering a more effective approach to balancing the trade-off between identity and text alignment.

**Weaknesses:**

- My main concern is comparison with new existing work. There have been several recent works after Textual inversion and Dreambooth that have significantly advanced this area. Some of them like SuTI (NeurIPS 2023), Instruct Imagen (CVPR 2024), HyperDreambooth (CVPR 2024) are also zero-shot which do not require any fine-tuning during evaluation.

- It is critical for this work to compare with the state-of-the-art papers in this area. I have listed a few above. However, I am sure that there could be more papers in the past two years in this area.

**Questions:**

- Comparison with multiple recent baselines is required. It would be great if the authors can provide this.

**Limitations:**

- While the proposed approach is novel, the core idea of decomposing the personalization process into multiple stages is not entirely groundbreaking. A more comprehensive exploration of the relationship between the proposed method and existing work would be beneficial.

---

> ### Author Rebuttal · Authors · 2024-08-07
>
> We would like to thank the reviewer for the constructive comments. We response the reviewer’s concerns as follows.
>
> >**W1: My main concern is comparison with new existing work. There have been several recent works after Textual inversion and Dreambooth. Some of them like SuTI (NeurIPS 2023), Instruct Imagen (CVPR 2024), HyperDreambooth (CVPR 2024).**
>
> We would like to clarify some aspects regarding the baseline methods described in our paper. As indicated in Figure 6 of the main paper and Figure 8 of the appendix, our comparisons are not limited to Textual Inversion and DreamBooth. We also include evaluations against other recent methods such as OFT (NeurIPS 2023) [1] and NeTI (SIGGRAPH Asia 2023) [2]. We adopted these two methods as baselines because they are among the state-of-the-art and are open-source.
>
> Thank you for the suggestion to compare our method with SuTI [3], Instruct-Imagen [4], and HyperDreamBooth [5]. In Figure R1 of the attached PDF, we present comparisons with SuTI and Instruct-Imagen. Due to the unavailability of open-source models for these two methods, we use the examples provided in their papers for comparison. As shown, our model achieves superior performance in text alignment compared to these methods. For instance, in the example of "[V] fancy boot with silver-tipped toes kicking a football", our model modifies only the toes to be silver, whereas SuTI modifies the entire boot. Regarding HyperDreamBooth, it focuses on personalizing human faces, whereas our approach targets general objects. Personalization of human faces is beyond the scope of our paper. We will include these comparisons in our revised paper.
>
> >**W2 and Q1: It is critical for this work to compare with the state-of-the-art papers in this area. I have listed a few above. However, I am sure that there could be more papers in the past two years in this area.**
>
> Thanks for the suggestion. As illustrated in the response to point W1, our current comparison includes two recent state-of-the-art methods: OFT (NeurIPS 2023) [1] and NeTI (SIGGRAPH Asia 2023) [2]. In addition to the above comparison to SuTI and Instruct-Imagen, we further include comparisons with two other open-source models, DreamMatcher (CVPR 2024) [6] and FreeCustom (CVPR 2024) [7], in Figure R2 of the attached PDF. As can be observed, our method achieves superior performance in both identity preservation and text alignment compared to these methods. We will add these comparisons to the revised paper.
>
> >**Limitation: While the proposed approach is novel, the core idea of decomposing the personalization process into multiple stages is not entirely groundbreaking. A more comprehensive exploration of the relationship between the proposed method and existing work would be beneficial.**
>
> The relationship between our method and existing multi-stage methods is discussed in the 'Multi-Stage Personalization' section of Related Work (i.e., Lines 95 - 108). Our method differs from existing methods in several aspects. Firstly, the motivation for our first stage (i.e., optimizing the textual embedding) differs from existing methods, where we focus on learning the embedding alignment while mitigating the risk of overfitting. Consequently, we significantly reduce the optimization steps to just 60 and lower the learning rate. Secondly, we decompose the learning process into three stages: learning the embedding alignment, refining the attention map, and capturing the subject identity. Thirdly, we utilize the super-category token to guide the attention map learning in a self-supervised manner throughout all training stages. We will include more multi-stage methods, such as MagiCapture [8], for comparison, making a more comprehensive discussion about multi-stage personalization methods.
>
> **References**
>
> [1] Zeju Qiu et. al. "Controlling Text-to-Image Diffusion by Orthogonal Finetuning". NeurIPS, 2023.
>
> [2] Yuval Alaluf et. al. "A Neural Space-Time Representation for Text-to-Image Personalization". SIGGRAPH Asia, 2023.
>
> [3] Wenhu Chen et. al. "Subject-driven Text-to-Image Generation via Apprenticeship Learning". NeurIPS, 2023.
>
> [4] Hexiang Hu et. al. "Instruct-Imagen: Image Generation with Multi-modal Instruction". CVPR, 2024.
>
> [5] Nataniel Ruiz et. al. "HyperDreamBooth: HyperNetworks for Fast Personalization of Text-to-Image Models". CVPR, 2024.
>
> [6] Jisu Nam et. al. "DreamMatcher: Appearance Matching Self-Attention for Semantically-Consistent Text-to-Image Personalization". CVPR, 2024.
>
> [7] Ganggui Ding et. al. "FreeCustom: Tuning-Free Customized Image Generation for Multi-Concept Composition". CVPR, 2024.
>
> [8] Junha Hyung et. al. "Magicapture: High-resolution multi-concept portrait customization". AAAI, 2024.

---

### Author Rebuttal · Authors · 2024-08-07

We would like to thank the reviewers for their constructive and thoughtful feedback. We are encouraged that the reviewers find our idea novel (ZpWY) and interesting (hyCc), and our method reasonable and clear (U1BU). We are pleased that they consider our results to be good (U1BU) and competitive compared to state-of-the-art approaches (hyCc). We appreciate their recognition of our paper as well-written and easy to understand (hyCc, FYJx). Moreover, ZpWY, U1BU, and FYJx recognize that our approach addresses a key problem in text-to-image personalization, while ZpWY confirms that our approach advances the field.

**[Response to a Common Concern]**

Below, we address a common concern regarding the training time of our model. Point-to-point responses are included as a reply to each reviewer.

**1. Concern regarding the training time of our model**

Indeed, computational overhead is a limitation of our method. To address this, we explored a simple yet effective strategy to reduce the training time. This involves increasing the learning rate while simultaneously decreasing both the training steps and the batch size for our third training stage, which is notably the most time-consuming phase. Specifically, the third stage of our original model performs 500 steps with a learning rate of 2e-6 and a batch size of 8. The fast version now completes training in just 200 steps with a learning rate of 1e-5 and a batch size of 4. This adjustment significantly reduces the training time from 20 minutes to 6 minutes on average. Interestingly, this fast version maintains performance comparable to our original model, likely because the first two stages provide a convenient starting point, allowing for a higher learning rate in the third stage. The qualitative evaluation is provided in Figure R4 of the attached PDF, and quantitative results are detailed in the table below. We observed that the fast version model performs very closely to the original model for short prompts (e.g., it even slightly outperforms the original model in the quantitative evaluation), but it slightly underperforms for complex prompts (e.g., the second and fourth examples in Figure R4).

| Methods              | Identity Preservation | Text Alignment | Training Time |
| :---------------- | :------: | :----: | :----: |
| NeTI [1]  	    | 0.6901  | 0.2522 |  13 minutes |
| Ours-fast       |  0.7268   | 0.2536 | 6 minutes |
| Ours-original |  0.7257   | 0.2532 | 20 minutes |

We thank the reviewers for this point, and will definitely include these results of this fast version model in the revised paper!

**[Additional Experimental Results]**

We summarize our additional experimental results below. Please refer to the attached PDF file for the figures. In the following, please note that Figure R* denotes the figure in the attached PDF.
1. **Comparison to SuTI [1] and Instruct-Imagen [2].** In Figure R1, we present a visual comparison with SuTI and Instruct-Imagen. Due to the unavailability of open-source models for these two methods, we compare with the examples provided in their papers.
2. **Comparison to DreamMatcher [3] and FreeCustom [4].** In Figure R2, we present a visual comparison with DreamMatcher and FreeCustom.
3. **Fast version of our model.** In Figure R4, we provide the results of our fast version model, which significantly reduces the training time from 20 minutes to 6 minutes.
4. **Quantitative ablation study.** The results of the quantitative ablation study are detailed in the response to reviewer U1BU's point W2.
5. **DreamBooth with more training steps.** In Figure R3, we provide the results of training DreamBooth with more training steps.
6. **Different settings of TI+DB.** In Figure R5, we present a visual comparison with two different settings for TI+DB: 1) first tuning the textual embedding and then the U-Net, and 2) first tuning the U-Net and then the textual embedding.
7. **Comparison to using TI or DB individually.** In Figure R5, we present a visual comparison to using TI or DB individually.
8. **Our model using SD1.5.** In Figure R6, we present the results of our model using SD1.5.
9. **Our model with the prior preservation loss.** In Figure R7, we present a visual comparison of our models with or without the prior preservation loss.

All the above additional experimental results will be added to the main text or appendix of the revised paper.

**References**

[1] Wenhu Chen et. al. "Subject-driven Text-to-Image Generation via Apprenticeship Learning". NeurIPS, 2023.

[2] Hexiang Hu et. al. "Instruct-Imagen: Image Generation with Multi-modal Instruction". CVPR, 2024.

[3] Jisu Nam et. al. "DreamMatcher: Appearance Matching Self-Attention for Semantically-Consistent Text-to-Image Personalization". CVPR, 2024.

[4] Ganggui Ding et. al. "FreeCustom: Tuning-Free Customized Image Generation for Multi-Concept Composition". CVPR, 2024.

---

### Decision · Program_Chairs · 2024-09-25

**Decision:**

Accept (poster)

**Comment:**

This paper proposes a new method for personalized high-quality image generation, called AttnDreamBooth.

Intially, reviewers pointed out some concerns such as less originality, ambigous description, lack of comparisons while respecting the importance of the tasks addressed and its method efficacy, with three positive scores and one rejection score.

In particular, AC carefully read R-ZpWY's comments considering his/her rejection score.
Then, AC found that most methods that the reviewer suggested to use for comparisons are publihsed after NeurIPS deadline or are not available.

Considering the situation and the contributions of this paper, AC thinks this paper is enough to be presented in NeurIPS, so recommends accepting this paper.

AC asks the authors reflect all important comments of reviewers in the cam-ready version.